# Age-Related Changes in Landing Mechanics in Elite Male Youth Soccer Players: A Longitudinal Study

**Michal Lehnert** [1],*\ , **Jakub Krejčí** [1]\ , **Miroslav Janura** [1] **and Mark De Ste Croix** [2]

1    Faculty of Physical Culture, Palacky University Olomouc, 771 11 Olomouc, Czech Republic;
     jakub.krejci@upol.cz (J.K.); miroslav.janura@upol.cz (M.J.)
2    School of Sport and Exercise, University of Gloucestershire, Gloucester GL50 2RH, UK;
     mdestecroix@glos.ac.uk
*    Correspondence: michal.lehnert@upol.cz; Tel.: +420-734-682-290

**Abstract:** The aim of this longitudinal observational study was to examine the age differences in jump landing kinematics in 13–17-year-old male soccer players. Landing technique was evaluated in three consecutive seasons in U14 ($n = 15$) and the U16 ($n = 10$) competition age groups using the Landing Error Scoring System (LESS). For the LESS, ANOVA revealed no significant interaction effect ($p = 0.81$, $\eta^2 = 0.009$) or main effect for groups ($p = 0.15$, $\eta^2 = 0.086$), but a significant year effect was observed ($p < 0.001$, $\eta^2 = 0.265$). The LESS score in the U14 group decreased significantly in the second year ($5.7 \pm 1.9$, $p = 0.006$, $d = 0.84$) and in the third year ($5.9 \pm 2.0$, $p = 0.020$, $d = 0.70$) compared to the first year ($7.1 \pm 1.7$). The LESS score in the U16 group decreased significantly in the second year ($5.1 \pm 0.9$ points, $p = 0.034$, $d = 0.77$) and in the third year ($4.9 \pm 1.4$, $p = 0.013$, $d = 0.92$) compared to the first year ($6.4 \pm 1.2$). This trend was supported by the results of the separate assessment of the sagittal plane joint displacement. These findings support previous limited findings from cross-sectional studies, which point to improved landing mechanics during maturation. However, the LESS scores in both groups indicate that players who were around and/or post-peak height velocity during the observed periods may be considered "at-risk", and suggest that preventive training programs should be introduced at earlier stages of players' development.

**Keywords:** growth; maturation; adolescence; injury; Landing Error Scoring System





## 1. Introduction

Reaching the top level of performance in sports, including soccer, requires long-term age-appropriate training. In this process, successive development of motor skill competencies and physical fitness through appropriate training and competition workload, which consider physical and psycho-social developmental changes, must be accentuated [1–3]. The optimization of this process requires minimizing the occurrence of injuries due to an acute injury risk as well as overtraining and overreaching.

Soccer is a team sport that has a high occurrence of injuries, with the most common injuries in both adult and youth soccer players reported in the lower limbs, specifically the hamstring, knee joint and ankle [4–7]. In the case of knee joint injuries in soccer, the highest risk is of anterior cruciate ligament (ACL) injury [4]. Although not so frequent, ACL injuries represent the most serious injury burden in terms of long-term consequences, including high re-injury risk, decreased knee function and degenerative changes [8]. Epidemiological studies show that ACL injuries in adolescent soccer players are mostly of a non-contact nature, and during soccer match-play and training, they usually occur during dynamic high-intensity movements [3,9–12].

Numerous modifiable and non-modifiable risk factors have been associated with non-contact ACL injuries [13,14], and it has been suggested that poor lower extremity mechanics of jump landing is one of the movement patterns for ACL injury, as it stresses

the passive ligament structures beyond their capacity [15–18]. Specifically, the magnitude of the mechanical load on passive structures and the risk of injury are affected by movements in the frontal, sagittal and transverse planes [19,20]. Especially, increased medial knee displacement in the frontal plane and valgus moments of the knee joint during the impact phase of jump-landing tasks are considered important potential predictors of ACL injury [21–24]. In the sagittal plane, injury risk decreases with greater angular joint displacement, which can be expressed by the magnitude of the change in the flexion angle in the hip, knee and ankle joints from the time preceding the initial contact of the foot with the surface to the time of maximum flexion, the position of the torso at the initial contact and changes in torso flexion [17,25]. Importantly, limited sagittal plane joint displacement (SPJD) during landing is associated with less mechanical energy absorption and greater ACL loading through quadriceps muscle actions [26,27] and increased frontal plane knee motion and moments [28]. Given that jumps and subsequent landings are common in soccer, and also given the trainability of this aspect in landing technique, consideration of this knowledge may help in LCA injury risk management in players. Other biomechanical influential risk factors are restricted ankle dorsiflexion range of motion, hip external rotations, asymmetries between the lower extremities and peak vertical ground reaction forces (pVGRF) [13,22,26,29,30]. Current research also indicates that restricted ankle dorsiflexion range of motion is associated with decreased SPJD in the ankle, knee and hip joints when landing [23,30]. Contrarily, increased ankle mobility may improve landing mechanics by increasing sagittal plane joint displacement due to a softer landing reducing pVGRF [24,26,31].

The incidence of ACL injury in youth players increases with age [3,4,32,33]. The increase in risk of injury is also related to periods of accelerated growth during adolescence [6,34,35]. In this context, adolescence is an important period in the development of specific motor abilities (e.g., running speed, jump height) that might be influenced by "adolescent awkwardness" [36]. From this perspective, monitoring jump-landing mechanics during adolescence, especially during peak height velocity (PHV) and following the age of PHV, could provide useful information, given that injury incidence rates are greatest during this period [35].

To date, only a few cross-sectional studies have examined the effect of growth and maturation on the landing mechanics in youth athletes, and their results are conflicting. In a recent study [37], in the sagittal plane, no significant differences were demonstrated between pre-PHV, circa-PHV or post-PHV players in joint kinematics from a Drop Vertical Jump test. However, in the Tuck Jump Assessment, the group of pre-PHV players achieved significantly higher values of flexion in the hip and knee joint at initial foot contact compared with older players. Significantly higher values were also observed for maximal flexion in the hip joint and for plantar flexion in the ankle at the initial foot contact in the pre-PHV group compared with post-PHV. In a cross-sectional study on youth soccer players aged 10–18 years [34], vertical jump height and absolute pVGRF from single leg jumping increased during maturation; however, relative to body weight, a significant difference was observed only on the left leg in circa versus post-PHV players. A trend of reduction in knee valgus with maturation on both legs was observed, but the between-group differences were only between post-PHV and pre-PHV players on the left leg.

Current knowledge indicates that appropriate landing mechanics may reduce ACL load during jump landing and decrease the risk of knee injury [26,28,38], and that movement patterns can be modified during growth and maturation in youth [34,38–41]. In this context, video-based movement screening can be used to identify potential ACL risk factors related to movement characteristics [13,16,30]. The "Landing Error Scoring System" (LESS) is a comprehensive, valid and reliable approach to clinical assessment of jump-landing biomechanics, and was developed by Padua et al. [17]. By means of this tool, jump-landing quality is assessed by analyzing the records of landing in the sagittal and frontal planes during the jump-landing screening task [16,42]. A study on elite youth soccer players [16] showed that the LESS could be used as a potential screening tool to determine ACL injury

risk. The authors also suggested that whereas a high LESS score may lack precision in identifying which athlete will sustain an ACL injury, the LESS score may be effective in separating athletes into high-risk and low-risk subgroups [16]. The key cut-off value suggested for high injury risk is five points [17].

In conclusion, although many studies have focused on the quality of neuromuscular control during jump landing, only a few studies have focused on the assessment of kinetic and kinematic injury risk factors during landing. Moreover, these studies were of a cross-sectional design; longitudinal studies on changes in landing mechanics in youth soccer players with high validity are missing. Such studies with longitudinal design can provide new information in terms of the changes in movement mechanics during growth and maturation in soccer players, which could be useful for injury prevention management and performance enhancement. Therefore, the aim of this longitudinal observational study was to examine the age differences in landing kinematics in 13–17-year-old elite male soccer players. The hypothesis was that the landing kinematics would be improved with the chronological age in regularly trained 13–17-year-old elite soccer players.

## 2. Materials and Methods

### 2.1. Participants

The present mixed-longitudinal study involved a group of 89 male youth soccer players who were recruited from a professional soccer club in the Czech Republic. The players pertained to two chronological competition age groups at the beginning of the study: U14y (*n* = 44) and U16y (*n* = 45). The players played in the highest national soccer league in their respective age categories and had at least 6 years of organized training experience. The inclusion criterion was the absence of a severe thigh and knee injury during the period six months prior to the measurement. The reasons for exclusion were non-participation in training sessions as a result of injury exceeding 4 weeks during the 3-year period of observation, absence in the measurement session and, predominantly, a change in club or release from the club. This resulted in the final sample for data analysis consisting of 25 players, including 15 players from the U14 age group (training age 6–7 years) who were students of the sports elementary school, and 10 players from the U16 age group (training age 8–9 years) who were students of different types of secondary school. The description of the players is shown in Table 1.

**Table 1.** Anthropometric characteristics of players.

|  | Year | U14 (*n* = 15) | U16 (*n* = 10) | *p* |
|---|---|---|---|---|
| Chronological age (years) | 1st | 13.3 ± 0.4 | 15.4 ± 0.4 | 0.31 |
|  | 2nd | 14.3 ± 0.4 | 16.4 ± 0.4 |  |
|  | 3rd | 15.2 ± 0.3 | 17.4 ± 0.3 |  |
| Maturity offset (years) | 1st | −0.3 ± 0.7 | 1.8 ± 0.5 | 0.023 |
|  | 2nd | 0.6 ± 0.7 | 2.8 ± 0.6 |  |
|  | 3rd | 1.2 ± 0.7 | 3.3 ± 0.6 |  |
| Stature (cm) | 1st | 161.2 ± 8.7 | 178.7 ± 3.7 | 0.023 |
|  | 2nd | 168.9 ± 9.4 | 181.9 ± 4.5 |  |
|  | 3rd | 171.9 ± 10.2 | 182.5 ± 4.2 |  |
| Body mass (kg) | 1st | 47.8 ± 9.6 | 67.2 ± 5.2 | 0.12 |
|  | 2nd | 57.8 ± 10.5 | 72.5 ± 4.6 |  |
|  | 3rd | 63.7 ± 12.0 | 75.7 ± 4.3 |  |

*n*—sample size; *p*—significance of the comparison between the U14 group in the 3rd year and the U16 group in the 1st year (Mann–Whitney U-test).

All of the participants trained five to seven times a week (total of 7–12 h) and usually played one competitive match in a week. Training typically consisted of age-related training aimed at youth athletes on an elite pathway: physical fitness (especially strength and power, speed and agility with and without a ball, repeated sprint ability with and without a ball) and skill-oriented training (technical–tactical training, game-like training,

recovery training). Neither age group in the current study was involved in systematic landing training, although plyometric training was included in the conditioning program of both groups.

The study was approved by the Ethics Committee of the Faculty of Physical Culture, Palacky University (approval no. 14/2015) and conformed to the Declaration of Helsinki regarding the use of human subjects. All of the players were informed about the objective of the research and the applicable testing procedures. The players' parents and the children submitted a written informed consent expressing their agreement with the testing procedure and the use of the data for further research purposes. The testing was performed over three competitive seasons, 2016/17, 2017/18 and 2018/19, and approximately three weeks into each season (end of August), always in the club's sports hall during the training units of the observed groups. The day prior to the testing, the participants were not exposed to any high-intensity exercises.

### 2.2. Procedures

#### 2.2.1. Anthropometrics

The date of birth and the date of the testing session at each time point were used for the determination of chronological age. The participants' biological maturity at each time point was predicted as the offset from PHV using the gender-appropriate equation according to Mirwald et al. [43]. For this purpose, leg length, tibia length and sitting height as well as standing measures were obtained using a stadiometer A-226 Anthropometer (Trystom, Olomouc, Czech Republic). A Tanita UM-075 weighing scale (Tanita, Japan) was used to measure body mass.

#### 2.2.2. Landing Error Score System

The study used the LESS, a 17-item scoring system devised by Padua et al. [17] that counts technique errors using a standardized checklist, to identify the players' abnormal jump landing kinematics and potential ACL injury risk. A higher LESS score is indicative of poor technique, while a lower score reflects better jump-landing technique. One group of items focuses on the position of the lower extremities and the trunk at the moment of the initial contact with the ground (items 1–6). The purpose of the second group of items is to assess errors in foot positioning (items 7–11) at initial ground contact (item 11), at the moment the entire foot is in contact with the ground (items 7 and 8) and between the initial contact and maximum knee flexion (items 9 and 10). The third group of items is focused on the movement of the lower extremities and the trunk between the initial ground contact and the moment of maximum knee flexion angle (items 12–14) or the moment of maximum knee valgus angle (item 15). The last two "global" items address the overall sagittal plane movement and the rater's general perception of landing quality (items 16 and 17) [43]. In addition to the overall LESS score, SPJD (item 16) was also separately analyzed to detect the players' stiff jump landing, as it substantially influences the pVGRF during landing and increases risk of ACL injury in players [13,44]. The LESS was previously reported to be a valid and reliable clinical tool to identify injury risk movement patterns occurring during jump landing, which enables the assessment of the risk of ACL injury in elite youth soccer players [16,17,39].

The LESS was applied to evaluate the execution of a single leg countermovement jump. The players performed four jump trials (one practice and three measured trials). The average value calculated from the three trials was used in the subsequent analysis [42]. The players were instructed to jump as high as possible off one leg (preferred) following a 2-step run-up, replicating heading a soccer ball to increase the ecological validity of the test [45], and to land on two feet. Additionally, two high-definition video cameras SONY HXR-MC2000 and SONY HXR-NX5E (SONY Corporation, Tokyo, Japan; frequency 25 Hz) were positioned on tripods two meters from the marked landing area in the frontal and sagittal plane to enable a two-dimensional biomechanical analysis. From the moment of the initial contact frame, which was defined as the frame immediately before the foot was flat

on the ground, the maximum flexion of the knee joint and the maximum valgus position were determined. The videos of all of the players were scored retrospectively by the same trained rater. For the evaluation of images, ImageJ software (National Institute of Health, Bethesda, MD, USA) and Kinovea 0.8.15 (Free software foundation, Boston, MA, USA) were used.

### 2.3. Statistical Analysis

The data are presented as means and standard deviations. The normal distribution was verified using the Kolmogorov–Smirnov test. The Mann–Whitney U-test was used to compare anthropometric variables between the U14 group in the 3rd year of measurement and the U16 group in the 1st year of measurement. The analysis of variance (ANOVA) for repeated measures was used to evaluate the changes in the LESS score and SPJD score. The following factors were considered for ANOVA: between-group factor (levels: U14 and U16), within-group year factor (levels: 1st, 2nd and 3rd year of measurement) and interaction factor. The within-subject standard deviation was calculated as the square root of the mean square error obtained from the ANOVA. Whenever the ANOVA revealed a significant factor, pairwise comparisons were performed using Fisher's post hoc test. For all statistical tests, $p < 0.05$ was considered significant. In addition to statistical significance, effect size measures were also used. Partial eta-squared ($\eta^2$) was used for ANOVA factor and Cohen's standardized difference ($d$) was used for pairwise comparison. Effect sizes were interpreted according to the following thresholds [46]: trivial ($\eta^2 < 0.01$, $d < 0.2$), small ($\eta^2 \geq 0.01$, $d \geq 0.2$), medium ($\eta^2 \geq 0.06$, $d \geq 0.5$) and large ($\eta^2 \geq 0.14$, $d \geq 0.8$). STATISTICA version 14.0 (TIBCO Software, Palo Alto, CA, USA) was used for statistical analyses. A sensitivity analysis was performed using G*Power version 3.1.9.6 (Heinrich-Heine-Universität, Düsseldorf, Germany). A two-sample two-tailed t-test, a significance level of 0.05, a power of 0.80 and a sample size of 10 were considered. This resulted in a detectable effect size expressed as a Cohen's $d$ of 1.00.

### 3. Results

The distributions of age ($p = 0.001$) and body mass ($p = 0.017$) were significantly different from the normal distribution. Therefore, the non-parametric Mann–Whitney U test was used to analyze all of the anthropometric variables. The anthropometric characteristics of the players in the U14 and the U16 competition age groups are shown in Table 1. The LESS score ($p = 0.083$) or the SPJD score ($p = 0.41$) were not significantly different from the normal distribution. Therefore, the parametric ANOVA was appropriate for the analysis of the abovementioned variables.

For the LESS (Figure 1a), ANOVA revealed a significant year factor ($p < 0.001$, $\eta^2 = 0.265$, large effect). Neither the group factor ($p = 0.15$, $\eta^2 = 0.086$, medium effect) nor interaction ($p = 0.81$, $\eta^2 = 0.009$, trivial effect) were significant. The within-subject standard deviation was 1.3 points. The post hoc analysis showed that the LESS score in the U14 group decreased significantly in the second year ($5.7 \pm 1.9$ points, $p = 0.006$, $d = 0.84$, large effect) and in the third year ($5.9 \pm 2.0$ points, $p = 0.020$, $d = 0.70$, medium effect) compared to the first year ($7.1 \pm 1.7$ points). The change between the second and third year was not significant ($p = 0.62$, $d = 0.14$, trivial effect). The LESS score in the U16 group decreased significantly in the second year ($5.1 \pm 0.9$ points, $p = 0.034$, $d = 0.77$, medium effect) and in the third year ($4.9 \pm 1.4$ points, $p = 0.013$, $d = 0.92$, large effect) compared with the first year ($6.4 \pm 1.2$ points). The change between the second and third year was not significant ($p = 0.69$, $d = 0.14$, trivial effect).

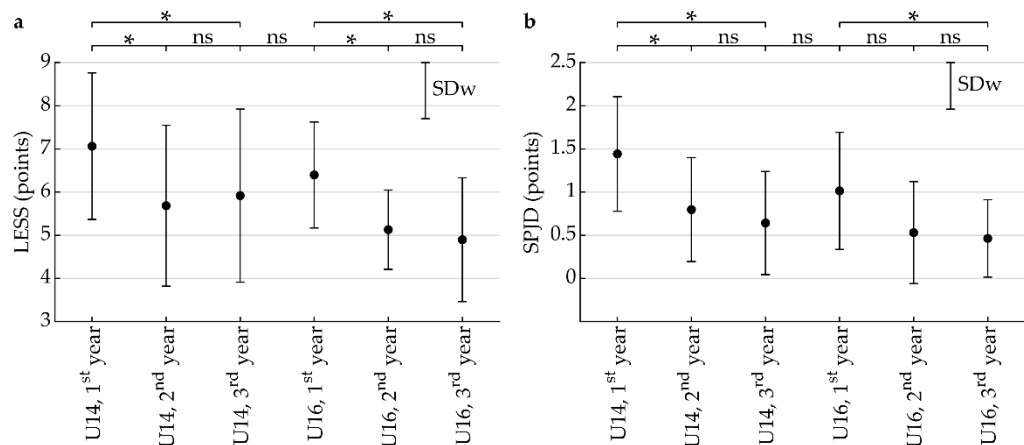

**Figure 1.** Age-related changes in Landing Error Scoring System (**a**) and sagittal plane joint displacement (**b**). Values are presented as arithmetic means and standard deviations. Only within-group pairwise comparisons are displayed. LESS—Landing Error Scoring System; SPJD—sagittal plane joint displacement; SDw—within-subject standard deviation; *—statistically significant ($p < 0.05$, Fisher's post hoc test); ns—not significant.

For SPJD (Figure 1b), the ANOVA revealed a significant year factor ($p < 0.001$, $\eta^2 = 0.320$, large effect). Neither the group factor ($p = 0.10$, $\eta^2 = 0.113$, medium effect) nor interaction ($p = 0.72$, $\eta^2 = 0.014$, small effect) were significant. The within-subject standard deviation was 0.5 points. The post hoc analysis showed that the SPJD score in the U14 group decreased significantly in the second year ($0.8 \pm 0.6$ points, $p = 0.002$, $d = 1.06$, large effect) and in the third year ($0.6 \pm 0.6$ points, $p < 0.001$, $d = 1.32$, large effect) compared to the first year ($1.4 \pm 0.7$ points). The change between the second and third year was not significant ($p = 0.43$, $d = 0.26$, small effect). The SPJD score in the U16 group decreased significantly in the third year ($0.5 \pm 0.4$ points, $p = 0.027$, $d = 0.91$, large effect) compared to the first year ($1.0 \pm 0.7$ points). The value in the second year ($0.5 \pm 0.6$ points) was not significantly different from either the first year ($1.0 \pm 0.7$ points, $p = 0.051$, $d = 0.80$, medium effect) or the third year ($0.5 \pm 0.4$ points, $p = 0.78$, $d = 0.11$, trivial effect).

## 4. Discussion

To our knowledge, this is the first longitudinal study exploring the age-related changes in the kinematic characteristics of jump landing in male adolescent soccer players based on field test data with suitable ecological validity. The main finding of this study is that the LESS score and SPJD score (as one from the items of the LESS scale) in both U14 and U16 competition age groups indicate improved landing techniques/mechanics with chronological age. The results also indicate that during nearly the whole observed period, the monitored elite youth players may be considered "at-risk". This suggestion, based on the longitudinal data, highlights the importance of attention to proper movement technique in players not only during the observed age groups but also during early stages of their pathway.

In the case of the LESS in both U14 and U16 group, the score was significantly higher in the first year of measurement compared with the second and third years (U14: chronological age 13 vs. 14 years and 13 vs. 15 years, respectively; U16: chronological age 15 vs. 16 years and 15 vs. 17 years, respectively). A significant difference in the LESS score was also recorded when comparing the results of the first year of measurement for the U14 category with the second and third years of measurement for the U16 category (chronological age 13 vs. 16 and 17 years, respectively). The improvement in the LESS score with age observed in this study might be explained by neuromuscular and structural changes occurring during growth and maturation, having a positive effect on the control of gross and fine motor skills and increase in muscle strength. Specifically, they deal with the improvement of muscle pre-activation, stretch reflex, rate of increase in electromyographic activity,

recruitment of motor units, reduction in muscle co-contraction and changes in tendon size and structure [36,47–50].

A comparison of the LESS score in the players of the same chronological age (14 years) in both groups (third year for the U14 group 5.9 ± 2.0 points; first year for the U16 group 6.4 ± 1.2 points) suggests a poorer landing technique in players from the U16 group. However, the difference (0.5 ± 1.3 points) was non-significant. When interpreting this difference, the intra-rater reliability of the LESS (standard error of measurement = 0.42) (20) must be taken into account. This difference may be attributed to the different quality of landing resulting from a degree of variability in the training process. Although the players in both groups were from the same club and participated in age-related training based on the recommendations of the Football Association of the Czech Republic, they were coached by different coaches, and it is obvious that there were some nuances both in the training content and the coaching approach.

A comparison of the results of this longitudinal study in the case of the LESS scale with the results of other studies is difficult, as we did not find a study that used the same or a similar comprehensive evaluation method to express changes in landing technique in adolescent athletes. However, in one of the previous studies, Padua et al. [16] reported an increased risk of non-contact injury in elite young soccer players who scored five or more points in the LESS compared with players with less than 5 points. Although the players in our study demonstrated positive age-related changes in landing technique, the average LESS score was greater than five points in each year of observation, except the third year of measurement for U16 players (4.9 ± 1.4 points), who can be considered post-PHV (maturity offset = 3.3 ± 0.6) [34]. This suggests that during nearly the whole observed period, players may be considered "at-risk". Padua et al. [16] also indicated that in elite youth soccer players aged 14 years, a LESS score of six or more was related to a greater risk of injury compared with individuals with a value of four or less. Values of 7.1 ± 1.7 in the U14 group in the first year of the study and 6.4 ± 1.2 in the U16 group in the first year of the study place these players above the cut-off suggested for low risk. Regarding the injury risk, it was also suggested that the period of PHV or post-PHV was an ACL rupture risk factor due to impaired knee joint mechanics when landing [51]. This suggestion is concerning for players in the U14 group who can be considered circa-PHV in the first year of observation (maturity offset = −0.3 ± 0.7 years) [34]. Poor landing mechanics at this developmental phase might be attributed to the phenomenon of "adolescent awkwardness", which is a temporary impairment of motor coordination explained by the rapid growth of bones during PHV not followed by equally rapid soft tissue adaptation. This discrepancy phenomenon results in reduced tendon muscle flexibility, leading to a significant reduction in joint range and the development of abnormal movement patterns and, consequently, increased susceptibility to lower limb injuries in soccer players [35,37,52]. On the other hand, it should be mentioned that no players monitored in this study suffered from an ACL injury during the observed period. It is assumed that the main reason was that the group of players in our study was small, and the incidence of ACL injury is not high and is lower in males compared with females.

Our findings concerning the poor landing technique in PHV players and gradual improvement during adolescence support the findings of a recent cross-sectional study on soccer players by Read et al. [34]. The authors observed 10–18-year-old soccer players and found that the decrease in pVGRF did not occur linearly with increasing chronological or biological age, but there was a temporary increase in landing stiffness during PHV. Unfortunately, the players in our study were not measured in the previous stage of maturation (i.e., pre-PHV period), and thus, it is not possible to compare the data from the first year with the data from the pre-PHV stage to confirm a potential decrease in landing technique around PHV. Similarly, the results of the current study indicate that although the players were involved in a systematic training process within a professional academy soccer club for at least six years at the beginning of the study, the training program applied in these years of elite developmental pathway did not effectively influence players' neuromuscular system

to assure better landing technique quality. As the movement pattern of jump landing pertains to the controllable risk factors for injury, and can be, to a large extent, modified, it is suggested that more attention should be paid to the quality of landing mechanics in the training process before PHV to improve players' landing mechanics and to decrease the incidence of ACL injury, especially in the subsequent "at-risk" periods. Research has shown that a properly designed 4-week neuromuscular training program reduced LESS scores and improved landing mechanics in both pre-PHV and post-PHV male cricket players [53].

The finding concerning improved jump-landing mechanics with age indicated by the changes in the LESS score in the soccer players observed is also supported by the decline in the SPJD score. Significantly lower values were found among players of the U14 competition age group in the second and third years of measurement compared with the first year. A positive but non-significant trend was also recorded when comparing the second and third years of measurement. In the U16 competition age group, significant improvements were shown only between the third year of measurement compared with the first year. Similarly to the LESS, the between-group comparison of the SPJD score showed significant differences when comparing the results of the first year of measurement for the U14 age category with the results of the second and third years of measurement for the U16 category. From the mechanistic perspective, the data point to a greater range of flexion movement in the key lower limb joints with age, which is associated with overall greater muscle work in order to stop the movement and consequently achieve softer and more efficient landings in players. In such landings, more mechanical energy can be absorbed, thereby reducing GRF and thus the load on passive joint stabilizers, especially ligaments [26,29,54]. As the magnitude of the pVGRF on landing after a jump can be more than ten times higher compared with stance, these changes may have a crucial role in the reduction in lower limb non-contact injury risk [55,56]. The ACL load during soft landing is also reduced by lowered anterior tibial forces produced by the contraction of the quadriceps. These forces at small knee flexion angles (0–30°) are associated with significant anterior tibial translation and increased LCA loading [27]. Reduced pVGRF, normalized to body weight, was also reported in a two-year longitudinal study on 13-year-old male and female adolescent basketball players who were classified as pubertal in the first year of the study and as post-pubertal in the second year of the study [57], as well as in previous cross-sectional studies in the general population [11,34,54,58]. On the contrary, improved landing mechanics with age in young soccer players were not confirmed in a recent cross-sectional study by Robles-Palazón et al. [37]. In this study, a sagittal tuck jump kinematic analysis of the landing showed higher values of flexion in the hip and knee joint at the initial contact of the legs in the pre-PHV group (<1 years) compared with the circa-PHV (−0.5 to 0.5 years) and post-PHV (>−1 years) groups. Significantly higher values were also reported for maximal flexion in the hip joint and for plantar flexion in the ankle at the initial foot contact in the pre-PHV group compared with post-PHV. Nevertheless, the comparison of the results of our study with these results is limited by the absence of a group of pre-PHV players in our study and also by the different division of players into groups (chronological age vs. biological age).

There are some limitations of the current study that should be considered. First, regarding the high fluctuation of players in the elite club and due to other reasons (illness, injury etc.), a higher number of players could not be included into the analysis. Second, the players were grouped based on competition age, while the maturity offset was assessed to characterize the maturity status of the players observed. Third, the design of the study did not allow us to identify the landing mechanics of the players in the pre-PHV phase, which would increase the scientific contribution and the application potential of the study. Fourth, it should be acknowledged that the study included players from a single elite club who were trained by different coaches. For this reason, the generalizability of the findings of this study to other youth athletes is limited.

## 5. Conclusions

The findings of the current longitudinal study on youth soccer players confirmed the results of the few cross-sectional studies both on athletic and non-athletic populations, which point to improved lower limb control of jump landing with age and maturation. However, even though the positive trend was evident in both the U14 and U16 competition age groups, the LESS scores during the observed period, except for players in the U16 group in the third year of observation, suggest that players who went through a systematic age-related training process may be considered "at-risk". The worse landing technique observed in the first year of the study in U14 players who were in or around PHV indicate that more attention should be paid to the quality of movement skills in youth soccer players during their elite pathway, and preventive training programs should be integrated at earlier stages of player development.

**Author Contributions:** Conceptualization, M.L. and M.D.S.C.; methodology, M.L., M.J. and M.D.S.C.; investigation, M.L. and M.J.; data curation, M.L. and J.K.; formal analysis, J.K. and M.J.; writing—original draft preparation, M.L., J.K., M.J. and M.D.S.C.; visualization, M.L. and J.K.; funding acquisition, M.L. All authors have read and agreed to the published version of the manuscript.

**Funding:** This research was funded by the Czech Science Foundation (GACR), grant number 16-13750S.

**Institutional Review Board Statement:** The study was conducted according to the guidelines of the Declaration of Helsinki and approved by the Ethics Committee of the Faculty of Physical Culture, Palacky University Olomouc, Olomouc, Czech Republic (no. 14/2015, 19 March 2015).

**Informed Consent Statement**Informed consent was obtained from all subjects involved in the study.

**Data Availability Statement:** The data presented in this study are available on request from the corresponding author. The data are not publicly available due to ethical and privacy restrictions.

**Conflicts of Interest:** The authors declare no conflict of interest.

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
