# Peer review of "Age-Related Changes in Landing Mechanics in Elite Male Youth Soccer Players: A Longitudinal Study"

_applsci, doi:10.3390/app12115324_

Round 1
Reviewer 1 Report
Dear Authors,
I have no particular concerns about the study presented in your manuscript. However, I believe that you should stress more what is the originality of your finding with respect to previous studies, correctly cited in the References.
Though your study is a longitudinal one, you should discuss how the LESS score 5.9+/-2 relative to the third year of study on the U14 group compares with the 6.4+/-1.2 relative to the first year of study on the U16 group, as both should refer to players of the same age (ca 15 yo) and training age.
Again on this point, Figure 1 shows in a graphical form the LESS scores as reported in the text: it does not add information, and visually the differences between the three years in each group are not of immediate impact, as the SD bars are relatively large. I would suggest to add a figure (or substitute Fig.1) with the average difference (and relative SD) in the LESS score obtained by the individuals during year 2 and 3 relative to year 1. This difference should be around 1, right?, and the SD smaller than 1, if the difference is significant.
Please check whether, in some points through the text, "LESS" should be changed to "LESS score".
Author Response
We would like to thank for the time taken to review the manuscript and comments which we feel have helped to improve the quality of the manuscript.

Reviewer 2 Report
STRUCTURE
- The manuscript is properly structured.
TITLE AND ABSTRACT
- The title or abstract should inform that the type of study.
INTRODUCTION
- The article does not include any pre-established hypotheses.
- It is recommended to start the introduction with the second paragraph and then the first paragraph.
- Provide current information about sagittal plane joint displacement (SPJD) because it is an outcome variable of the study.
- Howe L, S North J, Waldron M, Bampouras TM. Restrictions in Ankle Dorsiflexion Range of Motion Alter Landing Kinematics But Not Movement Strategy When Fatigued. J Sport Rehabil. 2021 Feb 11;30(6):911-919. doi: 10.1123/jsr.2020-0429. PMID: 33571960.
- Howe LP, Bampouras TM, North JS, Waldron M. Improved Ankle Mobility After a 4-Week Training Program Affects Landing Mechanics: A Randomized Controlled Trial. J Strength Cond Res. 2020 Jul 20. doi: 10.1519/JSC.0000000000003717. Epub ahead of print. PMID: 32694287.
- “Current knowledge indicates that appropriate landing mechanics may reduce ACL load during jump landing and decrease the risk of knee injury, and that movement patterns can be modified during growth and maturation in youth.” No reference is alluded to in support of this finding. Some of the recent research that would be interesting to include in this introduction are:
- Hanzlíková I, Hébert-Losier K. Is the Landing Error Scoring System Reliable and Valid? A Systematic Review. Sports Health. 2020 Mar/Apr;12(2):181-188. doi: 10.1177/1941738119886593. Epub 2020 Jan 21. PMID: 31961778; PMCID: PMC7040940.
- Lally EM, Ericksen H, Earl-Boehm J. Measurement Properties of Clinically Accessible Movement Assessment Tools for Analyzing Jump Landings: A Systematic Review. J Sport Rehabil. 2022 Jan 6:1-11. doi: 10.1123/jsr.2021-0288. Epub ahead of print. PMID: 34996030.
- Christopher R, Brandt C, Benjamin-Damon N. Systematic review of screening tools for common soccer injuries and their risk factors. S Afr J Physiother. 2021 Feb 12;77(1):1496. doi: 10.4102/sajp.v77i1.1496. PMID: 33824917; PMCID: PMC8010269.
MATERIAL AND METHODS
- It is recommended to make subsections organize the information such as: Study Design, Declarations: Ethics Approval, Consent to Participate and Consent to Publish, Eligibility Criteria, Study Intervention and Outcome Measures.
Study design
- The study does not specify the study used or how the participants were assigned. Describe the setting, locations, including periods of recruitment and data collect.
Ethic
- Indicate the code of ethics number and approval.
Anthropometrics
- Line 167: Do not use the first person plural. Applicable to the rest of the document.
Bias
- Specify the methods used to assess risk of bias in the study.
RESULTS
- Give characteristics of study participants (eg demographic, clinical, social) and information on exposures and potential confounders.
- Leg length, tibia length, standing and sitting height measurements are not shown in Table 1.
- Figure 1. Indicate the meaning of Landing Error Scoring System (LESS), as it has been done for SPJD.
- Line 240: is the first time this abbreviation SPJD is mentioned. Include the abbreviation the first time it is mentioned in the text (line 207).
DISCUSSION
- Provide a cautious overall interpretation of the results considering the objectives, multiplicity of analyses, results of similar studies, and other relevant evidence.
- Discuss the generalisability (external validity) and limitations of the study results.
REFERENCES
- References follow the indicated style.
- Many of the references are more than 5 years old, it is worth updating this research.
Author Response

(The authors gave the same response as above.)

Reviewer 3 Report
The purpose of the study was to explore age related changes in the kinematic characteristics of jump landing in male adolescent soccer players, in an ecological context.
The article is well written clear and with an interesting aim.
The introduction is well written and brings the reader to the main aim of the study. I do however suggest including a clear hypothesis.
Methodology:
I am not sure I understood correctly. You excluded from the longitudinal study those athletes that did not have an injury prior to the testing. Would those athletes not be a good sample to test as well? As a consequence, the present data show the LESS and SPJD scores only on the non injured athletes Is this not a bias?
Secondly, were all athletes from the same club?
Second, you affirm that the change in landing mechanics shown by the results is related only to the age of the athletes. You corroborate this hypothesis by reporting that the athletes were not “involved in systematic landing training” (line 136). If landing mechanics were only age-related, it’s presumable to expect the same LEES and SPJD values between the 3rd U14 group and 1st U16 group, where both athletes had the same chronological year value (15 years). Please provide this evidence and discuss.
Could you report some training data. How can you exclude a training effect on their improvement in the scores? This is an important missing part of the manuscript. At least volume/intensity and frequency of training increase throughout the 3 year period.
Line 190 – Justify the use of parametric analysis in such small sample size groups (10 and 15). We recommend carefully reading articles on the topic as Huta V. (2014) DOI: 10.20982/tqmp.10.1.p013 or Hopkins et al (2009) DOI: 10.1249/MSS.0b013e31818cb278 . Also, I suggest providing power analysis to determine the sample size and consider the use of non-parametric analysis tests.
Specific comments
Abstract
Line 11 - “specific periods of adolescence” but you do not indicate any real specific period throughout the abstract.
Line 13 – In the main text, both LESS, also the data about the SPJD are presented. Please provide this information also in the abstract.
Introduction
Line 63-91 – although giving detailed information about the current knowledge is always appreciated, too many details are provided in this section. I suggest reducing this part of the text to not overload the reader. You can provide this information in the discussion section (if suitable to the context)
Line 92-94 – please add a reference
Table 1 – Can you provide the anthropometric characteristic of the players also of the other years in which measurements were done?
Line 233 – the use of “players” instead of “participants” would be more accurate. Check the entire document
Line 139 – Provide the study’s code provided by the ethics committee.
Line 156 – eliminate the extra space
Line 167-168 Why a separate analysis of only SPJD was present? Please provide more information also in the introduction section about SPJD. Finally, consider utilizing the abbreviation SPJD in the main text and not only in Figure 1 b.
Line 189 – Please provide also detailed information and analysis of anthropometric characteristics of players of the following years. Are the 3rd year measurement of U14 comparable to 1st year U16?
Line 265…- A definition of when a LESS score can be considered as high or low, in relation to the risk of ACL injury occurrence, is presented only in this section. I suggest inserting this value’s interpretation also in the introduction and/or methods section.
Line 270-271 - a further discussion about the high LESS score found in the non injured (i.e. ) selected youth players is recommended.
Author Response

(The authors gave the same response as above.)

Round 2
Reviewer 2 Report
No further comments.
Author Response
Thank you for reviewing our manuscript.
Reviewer 3 Report
THe paper has greatly improved since the first version. However, I still have some methodological issues.
The introduction is still too long and does not provide sufficient information regarding why the authors chose to utilize the scores LESS and SPJD
I am still concerned about a possible training effect (therefore not only an age related effect) This point is poorly considered.
Line 11 – “10-17-year-old”, why 10? In the table the average age is 13 + 0.4
Line 16 – the data showed are not clear. For the whole abstract, consider to present LESS values and not only the statistical results .
Line 123-126 – Can you provide the exact number of players discarded for each reason?
Line 133 – add a space between “12” and “h”
Line 289-290 – I suggest to underline the fact that the players involved in the study were coached by different trainers and showed a different LESS and SPJD scores also in limitations and/or conclusion sections of the text. This fact could have a further discussion.
Author Response
We would like to thank you for reviewing the manuscript and for comments, which we tried to reflect. We made further reductions in the Introduction and at the same time, we added information regarding utilization the SPJD in our study. We also tried to explain the methodological issues, which were unclear, and extended limitations in this regards. In the abstract, LESS values are now presented as recommended.

Round 3
Reviewer 3 Report
THe authors did a great job in the revision process